# Novel Histopathological Biomarkers in Prostate Cancer: Implications and Perspectives

**DOI:** 10.3390/biomedicines11061552

**Published:** 2023-05-26

**Authors:** Paweł Kiełb, Kamil Kowalczyk, Adam Gurwin, Łukasz Nowak, Wojciech Krajewski, Roman Sosnowski, Tomasz Szydełko, Bartosz Małkiewicz

**Affiliations:** 1University Center of Excellence in Urology, Department of Minimally Invasive and Robotic Urology, Wrocław Medical University, 50-556 Wroclaw, Poland; kamil.kowalczyk@student.umw.edu.pl (K.K.); adam.gurwin@student.umw.edu.pl (A.G.); lukasz.nowak@student.umw.edu.pl (Ł.N.); wojciech.krajewski@umw.edu.pl (W.K.); tomasz.szydelko@umw.edu.pl (T.S.); 2Department of Urogenital Cancer, Maria Skłodowska-Curie National Research Institute of Oncology, 02-781 Warsaw, Poland; roman.sosnowski@gmail.com

**Keywords:** prostate cancer, prognostic markers, IL-17, STAT3, NRP1, LIMK1, Cofilin-1, CD169, PSMA, AMACR

## Abstract

Prostate cancer (PCa) is the second most frequently diagnosed cancer in men. Despite the significant progress in cancer diagnosis and treatment over the last few years, the approach to disease detection and therapy still does not include histopathological biomarkers. The dissemination of PCa is strictly related to the creation of a premetastatic niche, which can be detected by altered levels of specific biomarkers. To date, the risk factors for biochemical recurrence include lymph node status, prostate-specific antigen (PSA), PSA density (PSAD), body mass index (BMI), pathological Gleason score, seminal vesicle invasion, extraprostatic extension, and intraductal carcinoma. In the future, biomarkers might represent another prognostic factor, as discussed in many studies. In this review, we focus on histopathological biomarkers (particularly CD169 macrophages, neuropilin-1, cofilin-1, interleukin-17, signal transducer and activator of transcription protein 3 (STAT3), LIM domain kinase 1 (LIMK1), CD15, AMACR, prostate-specific membrane antigen (PSMA), Appl1, Sortilin, Syndecan-1, and p63) and their potential application in decision making regarding the prognosis and treatment of PCa patients. We refer to studies that found a correlation between the levels of biomarkers and tumor characteristics as well as clinical outcomes. We also hypothesize about the potential use of histopathological markers as a target for novel immunotherapeutic drugs or targeted radionuclide therapy, which may be used as adjuvant therapy in the future.

## 1. Introduction

Prostate cancer (PCa) is the second most commonly diagnosed cancer in men worldwide (with 1,414,259 new cases in 2020) and the seventh most frequent cause of death due to cancer, accounting for 375,304 deaths in 2020 [1]. Since life expectancy is increasing worldwide and PCa incidence is correlated with increasing age, we expect a rise in the number of men newly diagnosed with this type of cancer in the near future [2]. Fortunately, cancer-specific survival (CSS) in patients with PCa has also increased in the past few years. This is probably due to the development of more effective tools for screening and diagnosing PCa at an early stage and improved protocols for adjuvant therapy. However, despite the advances in adjuvant therapy, which treatment would be most beneficial for a given patient remains unknown. Moreover, there is still a lack of tools for predicting survival prognosis and the risk of cancer dissemination in patients after prostatectomy. Tissue markers represent tumor heterogeneity, which in clinical practice denotes different responses to certain types of adjuvant therapy, which we will further describe in this review. Currently, there are no guidelines recommending the use of markers in decision-making regarding treatment. However, in the future, incorporating these additional data into treatment may establish the foundation for a more personalized and effective approach to treating PCa patients. They could also serve as an additional prognostic factor to be used alongside clinicopathologic parameters, such as prostate-specific antigen level, histological grade group, and clinical stage. In this review, we focus on specific histopathological markers, particularly CD169 macrophages, neuropilin-1, cofilin-1, interleukin-17, signal transducer and activator of transcription protein 3 (STAT3), LIM domain kinase 1 (LIMK1), prostate-specific membrane antigen (PSMA), Appl1, Sortilin, Syndecan-1, AMACR, CD15, and p63, which in the future may help to establish the prognosis of patients with PCa as well as assist in choosing the most beneficial adjuvant therapy.

## 2. Evidence Acquisition

We used the PUBMED/Scopus database and gray literature to conduct a thorough search for original and review articles published up to December 2022 for the purposes of this narrative review. The search was limited to the English-language literature. The following terms were combined in our search: prostate cancer; biomarkers; IL-17; STAT3; NRP1; LIMK1; Cofilin-1; PSMA; AMACR; CD15; Appl1; Sortilin; Syndecan-1, and p63. Ultimately, 275 articles were selected for this review. Based on the authors’ agreement, 211 studies with the strongest level of evidence and relevance to the subject were selected.

## 3. Biomarkers

Histopathological markers can be roughly divided into two groups according to their localization. Extracellular biomarkers comprise surface antigens involved in the identification of cell type as well as cell differentiation (CD antigens) and transmembrane receptors acting as signal transducers, i.e., for vascularization, which is crucial in tumor progression. Intracellular biomarkers include transcription activators, cytoskeleton-bind proteins, intracellular receptors, cytokines, etc.

In Table 1, we summarize the characteristics of the novel biomarkers described in this review.

### 3.1. Extracellular Biomarkers

#### 3.1.1. CD169

One promising marker is the surface antigen CD169, which can be found on macrophages located in the sinuses of the lymph nodes. Various cells residing in the lymphatic system are known to be involved in anti-tumor activity, playing a pivotal role in the immune response against cancer cells. Macrophages found in regional lymph nodes absorb tumor-derived antigens, which are then presented to lymphocytes, including CD8+ T cells and natural killer (NK) cells responsible for cytotoxic responses. It has been suggested that CD169 is significant not only in PCa but also in melanoma, bladder cancer, endometrial tumors, and colorectal tumors [3,4]. The anti-tumor activity of these specific lymph node sinus macrophages has already been shown in animal model studies. Komohara et al. demonstrated that in colorectal and endometrial tumors, a high density of CD169 macrophages is associated with a higher number of infiltrating CD8-positive lymphocytes and NK cells responsible for a direct cytotoxic response. Further, a high number of CD169-positive macrophages in the tumor environment is also correlated with better overall survival and is independent of gender or age [3]. Some highly metastatic cancers are suspected of producing specific factors that precondition the tumor environment to promote their growth and metastasis. Such factors may potentially suppress the activity of macrophages via antigen CD169 expression in tumor-draining lymph nodes. Asano et al. showed that, in patients with bladder cancer, the abundance of CD169+ macrophages is correlated with a low T stage and a higher number of cytotoxic CD8+ cells. Moreover, the group with a high CD169 score had longer cancer-specific survival (five-year cancer-specific survival rate: 83.3% vs. 31.3%) [5]. Strömvall et al. found that PCa patients with low CD169 immunostaining in metastasis-free regional lymph nodes had worse outcomes (significantly shorter survival time) than patients with high CD169 scores—200 (95% CI 178–221) vs. 232 (95% CI 226–239) months [4]. However, the CD169 scores were not related to PSA relapse, although they were associated with an increased risk of death from PCa [4]. Figure 1 shows the proposed mechanism underlying the anticancer activity of CD169 macrophages.

#### 3.1.2. Neuropilin-1 (NRP1)

Another example of an extracellular biomarker is neuropilin-1 (NRP1), which acts as a transmembrane co-receptor and interacts with vascular endothelial growth factor (VEGF) [6]. The tumor microenvironment is distinguished by enhanced metabolic activity, which results in hypoxia-triggering hypoxia-inducible factors (HIFs). VEGF is one of the transcripts controlled by HIF-1 that supports each phase of the vessel formation cascade. HIF-1 consists of two different subunits, α and β [7]. When the oxygen level in tissue is not sufficient, subunit α of HIF-1 remains in its non-hydroxylated form inducing numerous gene products that act proangiogenic [8]. Interestingly accumulation of non-hydroxylated HIF-1α does not only occurs under extended exposure to hypoxic conditions. Numerous DNA mutations in cancer cells may result in genetic modifications, which hinder the ubiquitination and proteasomal degradation of HIF-1α [9,10]. Due to angiogenesis promotion, VEGF enables tissues to adapt to a hypoxic environment. By interacting as a transmembrane co-receptor with VEGF, NRP1 seems to be crucial for vessel development. Several studies have demonstrated that neuropilin is elevated in tumors such as breast cancer, neuroblastoma, colon cancer, and lung cancer, highlighting its role in neovascularization [11,12,13,14]. However, NRP1 overexpression in tumor cells is associated not only with angiogenesis but also with PCa cells developing resistance to androgen-targeted therapy and progression to metastatic castration-resistant prostate cancer (mCRPC). This information may help some patients, after radical prostatectomy, to avoid potentially inefficacious androgen-targeted therapy and to choose more effective options for the management of the disease. Tse et al. demonstrated that NRP1 expression in cancer tissue is positively correlated with increased Gleason grade and pathological T scores, positive lymph node status, and the failure of primary therapy. Flow cytometric tests have found rather low NRP1 levels in benign prostate epithelial cell lines and demonstrated elevated NRP1 levels in tumor cells with increased metastatic potential, with the highest expression found in mCRPC cells. Furthermore, one study reported a significantly lower chance of relapse-free survival in patients with NRP1 overexpression [15]. NRP1 may also serve as a predictor of adjuvant radiotherapy failure. Patients diagnosed with pT3 or positive margin disease with higher levels of the neuropilin-1 co-receptor were found to develop a post-RT biochemical recurrence (BCR) more often than those with lower marker expression. Furthermore, multivariate analysis has revealed high NRP1 expression to be an independent predictor of BCR. Finally, after radical prostatectomy, higher levels of the NRP1 marker are present in patients with distant metastasis than no metastasis, establishing NRP1 expression as a significant independent predictor of metastatic progression [15]. It is worth noting that NRP1 is expressed not only on cancer cells but also on immune cells such as regulatory T cells. Its expression levels appear correlated with the chemotherapy response in cervical cancer patients, as highlighted in a study by Battaglia et al., who observed a reduced number of regulatory T cell lymphocytes expressing NRP1 after neoadjuvant therapy (platinum-based CR) [16]. Moreover, decreases in NRP1 levels were directly correlated with tumor mass reduction. The authors hypothesized that this decrease might have been caused by stress signals released from disintegrating tumor cells, leading to the differentiation of regulatory/suppressor T cells into effector types. Interestingly, NRP1 seems to be not only a biomarker, which may help identify post-RP patients at risk of metastasis but also a potential target of immunotherapeutic agents. The idea of targeting the neuropilin-1 receptor has already been highlighted in two studies, which found that NRP1 blockade suppresses angiogenesis and medulloblastoma regression, decreases the rate of metastasis, and leads to better oncological outcomes in treated mice [17,18].

#### 3.1.3. CD15 (Lewis X/Lex)

Lastly, extracellular biomarkers are CD15 (Lewis X/Lex) and CD15s (a sialyl form of CD15), which are the fucosyl (3-fucosily-N-acetyl-lactosamine) moieties found on the surface of various cancer cells. These malignancies include gastrointestinal system cancers, breast cancer, hematological malignancies, lung cancer, gliomas, and melanoma [19,20,21,22,23,24,25]. The expression of CD15 and CD15s has also been confirmed in urological malignancies. These antigens have been demonstrated to occur on the surfaces of the bladder, renal, and PCa cells, while they are not usually detected in healthy tissue, except occasionally in umbrella cells [26,27,28,29]. In fact, in bladder and renal cancer, the diagnostic and prognostic value of CD15 and CD15s is not unequivocal [30]. Nevertheless, their value seems to be clearer in PCa, in which overexpression of these antigens had already been demonstrated in the 1990s [28,31,32]. The upregulation of CD15s is associated with hormonal resistance, aggressive disease, and poor prognosis [28,32]. The high expression of CD15 and CD15s may influence PCa progression through several mechanisms. First, CD15 and CD15s play a crucial role in the adhesion of cancer cells to the blood vessel endothelium [33,34,35,36]. The initial step of cell adhesion (tethering) between CD15s and selectins, a family of cell adhesion molecules located on the surface of endothelial cells, is essential for cancer metastasis [37,38]. Second, CD15 and CD15s can change the structure of prostate cell mucins (membrane-bound proteins) that enable cancer cells to hide from destructive NK cells [39]. Third, it has been established that CD15s is the specific ligand of leukocyte-expressed L-selectin [40] that allows the binding of cancer cells and leukocytes, facilitating the spread of metastases through the circulatory system. Recently, Munkley et al. found CD15s expression on PCa cells to be regulated by androgens [41]. This might be a valuable factor that explains why androgens play a crucial role in the development and progression of PCa and why androgen deprivation therapy (ADT) is usually the first-line treatment in metastatic disease. ADT affects the CD15s antigen via the modulation of androgen receptors (AR). The latter are responsible for regulating the biosynthesis of related glycans, such as CD15s, and it has been demonstrated that ADT reduces glycan production associated with PCa progression. Furthermore, a study used the newly developed anticancer drug compound 1 on androgen-independent PCa cell lines with the overexpression of CD15s on their surfaces [42]. The percentage of CD15s-positive cells was significantly lower after treatment. Targeting CD15 and CD15s in PCa treatment deserves attention and should therefore be further analyzed by researchers. CD15 and CD15s antigens were also recently observed on the surface of PSA from cancer cells [43]. The authors suggested that the CD15s expressed on PSA is a good candidate for a prognostic marker.

### 3.2. Intracellular Biomarkers

#### 3.2.1. Interleukin 17 (IL-17)

The IL-17 family is an example of a very broad intracellular biomarker group whose precise mechanisms of action and clinical application in cancer disease are still not fully known. The Il-17 cytokine family consists of six ligands (IL-17A to IL-17F) and five receptors (IL-17RA to IL-17RE). In most studies, the ligands IL-17A and IL-17E were investigated due to their unique biological functions in tumors [44,45]. IL-17F acts similarly to IL-17A, and studies have shown that these two ligands are crucial proinflammatory cytokines in inflammatory and autoimmune diseases [46,47]. Il-17E is an important cytokine in the pathogenesis of asthma and atopic diseases [48]. Cytokines of the IL-17 family have been reported to show ambiguous effects on tumor development: with some ligands showing pro- and others anti-tumor effects [49]. The studies to date show that IL-17 may promote cancer development in several types of non-urological cancers, such as colon cancer [50,51,52,53], skin cancer [54,55], lung cancer [56,57], and breast cancer [58]. The role of Il-17 in renal cancer development is primarily pro-tumor, but further investigation is required to determine its ultimate character [59]. The results of a complex study on the immunoreactivity of the ligands and receptors of the IL-17 family in patients with bladder cancer showed significantly elevated IL-17A, IL-17F, and IL-17RC immunoreactivity in the bladder cancer group but decreased IL-17E, IL-17RA, and IL-17RB immunoreactivity. The researchers concluded that changed patterns of the expression of IL-17 cytokine family ligands and receptors may contribute to the occurrence and development of bladder cancer [60]. The role of IL-17 in PCa development is still unclear. Elevated levels of mRNAs encoding IL-17A and IL-17RA were detected in PCa and benign prostate hyperplasia (BPH) tissue [61,62]. A more detailed study investigated the expression of a wide spectrum of IL-17 types and their receptors with a comparison between normal prostate, PCa, and BPH tissues, demonstrating that IL-17A, acting through the IL-17RA receptor, may contribute to the development of PCa and BPH. In contrast, the interaction of IL-17E with the IL-17RB receptor demonstrated an anti-tumor effect [63]. Other studies have shown that the expression of IL-17RC is significantly increased in castrate-resistant prostate cancer (CRPC) compared with hormone-sensitive prostate cancer [64,65]. Results from studies on mouse models suggest that IL-17 promotes the growth of prostate adenocarcinoma, even in castrate conditions [66,67], and may contribute to the metastasizing process [68]. Using a mouse model, Zang et al. described the molecular mechanism underlying the pro-tumor effect of IL-17 [69]. They designed their study based on the known phenomenon of the increased IL-17 and matrix metalloproteinase 7 (MMP7) expression in PCa tissue. The results of their study showed that IL-17 promotes prostate carcinogenesis through MMP7, which induces the epithelial-to-mesenchymal transition (EMT). In conclusion, the authors indicated that the IL-17–MMP7–EMT axis is a potential target of new PCa treatments [69]. A schematic of this mechanism is presented in Figure 2.

In one of the most recent studies, Janiczek et al. investigated the expression of IL-17 in the prostates of patients treated for PCa (including Gleason score stratification) and BPH. The findings of this study showed that IL-17A and IL-17F, acting through the IL-17RC receptor, were involved in the pathogenesis of PCa and BPH. Moreover, compared with IL-17A, expression of IL-17F was more often observed in PCa with a higher histological grade. Expression of the IL-17RA receptor was not detected in either PCa or BPH tissue. The authors concluded that the inflammatory process was more severe in BPH than in PCa. Additionally, the authors observed that a lower Gleason score for PCa was associated with higher expression of selected IL-17 types. These findings may suggest that the inflammatory process is more important in the carcinogenesis of lower grade PCa [70]. Although the role of the IL-17 cytokine family and their receptors in the pathogenesis of PCa currently remains unclear, the results of recent studies are promising. Future studies that investigate the expression of IL-17 in PCa may assist with determining the role of inflammation in carcinogenesis and, potentially, future immunotherapy.

#### 3.2.2. Cofilin-1 (CFL1)

Another mechanism in which intracellular biomarkers are involved is cancer cell migration and its ability to invade adjacent stroma as well as distant tissues. This process is reliant on constant cytoskeleton remodeling, which is regulated, inter alia, by cofilin-1 (CFL1), a protein that binds actin monomers and plays a key role in pseudopod formation and cytokinesis of cells [71]. CFL1 overexpression has been linked with the particular aggressiveness and migration rates of not only PCa cells but also breast, ovarian, and colorectal cancer cells [72,73,74,75,76]. It seems that CFL1 expression also has an impact on cancer cell resistance to systemic therapy. In a study by Xiao et al., knocking down the protein expression of cofilin-1 in PCa cells enhanced the anticancer effect of docetaxel [71,72]. Furthermore, docetaxel itself suppressed the expression of CFL1 in cancer cells [71]. In the future, this protein may serve as a target of immunotherapy prior to the administration of chemotherapy. Apart from its association with metastasis and chemoresistance, CFL1 is also associated with the clinicopathological characteristics of PCa. In a study by Lu et al., significantly higher expression of cofilin-1 (50% vs. 86.9%) was observed in patients with post-operative Gleason scores (GS) ≥ 7 than those with GS < 7, and CFL1 was considered an independent predictor of high GS. Furthermore, the increased expression of CFL1 was also observed in patients with lymph node metastasis (100 vs. 62.9%), although cofilin-1 status was independent of PSA level or age. The important takeaway from the study is that CFL1 seems to be specific to tissue samples positive for PCa: it was expressed in 70.3% of PCa specimens but not in any samples from the control group (patients with BPH) [77]. Chen et al. demonstrated, in their study [78], that cofilin-1 promoted cytoskeleton remodeling and activates the migration and invasion of PCa cells, significantly contributing to their metastatic potential. In the same study, the researchers found that cofilin-1 overexpression was linked with higher resistance to adriamycin, which was evidenced by higher IC50 values of the drug being found in CFL1 (+) cells [78].

#### 3.2.3. STAT3

Intracellular biomarkers are also important agents in the development of the pre-metastatic niche, and one of their most important representatives is the signal transducer and activator of transcription protein 3 (STAT3). The important role of STAT3 in tumor development has been well described [79]. Activated STAT3 in tumor cells is a crucial mediator of oncogenesis [79,80,81,82,83,84,85,86] and the cancer-related inflammation process. Active STAT3 promotes tumor cell proliferation, angiogenesis, immunosuppression, and tumor invasion; all of these factors create a pro-carcinogenic microenvironment [87] and facilitate inflammation. Via this process, a pre-metastatic niche is created, changing the environment to one of a future metastatic site for cancer cells [82,88]. Further, studies have shown that STAT3 may promote tumor growth and hamper the effects of treatment by influencing mitochondrial metabolism [89] or epigenetic regulation [90] or by inducing drug resistance [91,92]. Due to its important role in tumor development, STAT3 has been investigated as a potential therapeutic target for new drugs, and the results are promising [93,94,95,96,97]. The role of STAT3 overexpression in tumor tissue remains inconclusive. In patients with PCa [98], as well as with some other urological [99] and non-urological solid tumors such as ovarian cancer [100], hepatocellular cancer [101,102], pancreatic cancer [103,104], and renal cell carcinoma [105], STAT3 overexpression is linked with poor survival. Regarding patients with colorectal cancer [106,107,108], lung cancer [109,110,111,112,113], gastric cancer [114,115,116,117], and melanoma [118,119], the results of studies are ambiguous—STAT3 overexpression is found to be a factor of favorable prognosis in some while being related to poor outcomes in others. To date, research has unequivocally shown that STAT3 overexpression in breast cancer [120,121,122] tissue may be beneficial in terms of patient outcomes. In the most comprehensive meta-analysis to date, conducted by Pin Wu et al. [123], the authors evaluated the prognostic value of STAT3 expression (for different phosphorylation states) and its correlation with the clinical outcomes of patients with solid tumors, including PCa. The authors found that STAT3 overexpression was associated with worse three- and five-year overall and disease-free survival. Elevated expression of STAT3 (especially its phosphorylated form, pSTAT3) in PCa, similarly to most investigated solid tumors, was found to be associated with poor prognosis, with the exception of a better prognosis found only in the case of breast cancer. Additionally, a meta-analysis on STAT3 expression in PCa, conducted by Tam L. et al. in 2007 [98], suggested that the activated STAT3 pathway might induce progression to the hormone-refractory type of PCa and, simultaneously, that it was a potential target for new drugs.

#### 3.2.4. LIM Domain Kinase 1 (LIMK1)

Another intracellular biomarker involved in cancer cell migration, the metastatic process, and androgenic signaling is LIM domain kinase 1 (LIMK1). LIMK1 belongs to the LIM kinase protein family, and it is particularly involved in reorganization of the actin cytoskeleton, which is crucial for cell migration and metastasis [124,125]. Cytoskeleton remodeling is made possible by phosphorylation and inactivation of the cofilin that binds actin [126]. Further, LIMK1 seems to also be involved in intracellular androgen receptor signaling, which is especially promising in terms of developing new therapies targeting AR in CRPC patients, which could serve as alternatives to docetaxel [127]. This may contribute to avoiding the severe side effects of docetaxel therapy, such as cytopenia and pneumotoxicity. While reports on LIM domain kinase 1 (LIMK1) in PCa specimens are limited, it may be a promising marker, as in many other cancers are LIMK1-positive, including colorectal cancer [128], lung cancer [129], osteosarcoma [130], and breast cancer [131], which also has clinical implications. In colorectal cancer tissue specimens, LIMK1 upregulation was found to be associated with lower overall survival rates as well as increased lymph node metastasis potential [126]. In a study by Huang et al. [132], which strictly focused on PCa patients, LIMK1 expression was linked with worse clinicopathological characteristics as well as worse disease dissemination and oncological outcomes. The authors compared LIMK1 upregulation between PCa specimens and a control group of BPH tissues (77.1% vs. 26.0%). Interestingly, LIMK1 staining was higher in lymph node metastasis samples than in PCa tissue collected from the same patient. Furthermore, high LIMK1 expression was correlated with prostate volume, PSA, PSA density, Gleason score, T stage, lymph node metastases, extracapsular extension, seminal vesicle invasion, and positive surgical margins, though no association with patient age was found. Finally, in the multivariate analysis, an elevated LIMK1 level was an independent risk factor for lymph node metastasis and biochemical recurrence after prostatectomy. The idea of targeting LIMK1 in treating PCa patients was comprehensively investigated in a study by Mardilovich et al. [127]. In their study, a LIMK selective small molecule inhibitor (LIMKi) was used as a potential agent against PCa cells. LIMKi had a negative effect on cell movement as well as proliferation. Further, LIMKi increased the percentage of PCa cells with sub-G1 DNA content, which is an indicator of apoptosis. Interestingly, these effects were more pronounced in androgen-dependent than androgen-independent cells. LIMKi led to a reduction in AR nuclear transport and transcriptional activity, eventually causing decreased cell proliferation. Furthermore, the authors found clinical implications of LIMK1 expression in PCa specimens after radical prostatectomy: non-metastatic patients with high levels of LIMK1 presented significantly worse survival compared with the metastatic group. Additionally, the increased LIMK1 staining was correlated with lymphovascular invasion, which is considered an independent prognostic factor for biochemical recurrence [133] and is connected with poor outcomes [134]. However, no association between LIMK1 upregulation and patient age, Gleason score, or PSA level was found. In general, the findings suggested that LIMK1 regulateds AR function, leading to disease progression in AR-dependent cells and that it was a promising target for novel drugs, especially those used in the treatment of CRPC patients with increased intracellular AR activity [135].

#### 3.2.5. AMACR

The tumor microenvironment is characterized by the increased generation of reactive oxygen species (ROS) and a disturbed redox balance [136], which lead to DNA damage in cells by peroxides. One of the enzymes involved in this process is alpha-methylacyl-CoA racemase (AMACR), which is an enzyme that plays a significant role in lipid and drug metabolism. It is found in peroxisomes and mitochondria and is involved in the b-oxidation-mediated degradation of branched fatty acids [137,138]. Elevated AMACR levels have been linked to many cancer types, including renal cell carcinoma [139], gastric cancer [140], ovarian cancer [141], and hepatocellular carcinoma [142]. Combined with high molecular weight cytokeratin (HMW-CK) and p63, AMACR can also be used in histopathology staining as a negative marker of a benign prostate gland [143]. Its sensitivity as a prostatic adenocarcinoma-specific marker has been found to range from 80 to 100% [144,145,146,147,148]. It can be used together with positive markers for basal cells in prostatic glands, such as HMW-CK and p63, to increase the level of confidence in establishing a final diagnosis [149]. A comprehensive meta-analysis [150], which included 22 studies with 4385 participants, linked the positive expression of AMACR in prostatic tissue with the increased diagnosis of PCa (OR = 76.08; 95% CI, 25.53–226.68; P, 0.00001). The meta-analysis comprised studies from various geographic regions, demonstrating that marker expression in cancer did not differ significantly between Asians and Caucasians. However, there is no evidence for a statistically significant correlation between AMACR positivity and the Gleason score [149,151]. Since the first reports on its possible role as a diagnostic marker [152], AMACR has been thoroughly examined as a novel drug target [153,154]. Shan Zha et al. reported a fourfold increase in AMACR enzymatic activity in PCa cells compared with normal prostate cells. Small interference RNA (siRNA) knockdown of AMACR disturbed the proliferation of the androgen-responsive PCa cell line. In that same study, the authors also studied a treatment involving the combination of an AMACR inhibitor with anti-androgen therapy, which resulted in increased cell growth inhibition compared to treatment with either in isolation [152]. Interestingly, AMACR also plays a significant role in the progression of PCa from a hormone-sensitive to a hormone-refractory state. Although acquiring resistance to hormonotherapy is a complex process, Takahara et al. found much lower expression of AMACR in androgen-independent than in androgen-dependent cell lines. Furthermore, inhibiting AMACR expression using siRNA resulted in the increased expression of the androgen receptor as well as the decreased expression of insulin-like growth factor I and platelet-derived growth factor alpha (growth factors that regulate cell proliferation and blood vessel formation), which results in the reduced viability of cancer cells in androgen-depleted serum compared with untreated cells. The authors hypothesized that future therapies focusing on AMACR inhibition could theoretically convert PCa cells from being hormone-independent to hormone-dependent [155,156]. AMACR has also been evaluated as a novel diagnostic marker to detect PCa during screening tests. In a recent study by Xin et al. [157], the authors, after conducting digital rectal examinations, collected first-catch urine samples from patients who also underwent prostate biopsy. The urinary exosomal AMACR (UE-A) was then measured using ELISA. The authors concluded that the level of the examined marker was higher in PCa and clinically significant PCa (csPCa)—Gleason score ≥ 7—than in BPH (*p* < 0.001). Furthermore, UE-A helped to differentiate between PCa and BPH and between BPH plus non-significant prostate cancer (nsPCa) and csPCa with area under the ROC curve (AUC) values of 0.832 and 0.78 obtained, respectively. Testing UE-A was also superior to PSA (*p* = 0.0054), PSAD (*p* = 0.008), and f/t PSA (*p* = 0.056) in distinguishing PCa from BPH. The clinical utility of UE-A was also evaluated in a multi-center cohort of patients at initial biopsy: by establishing 95% sensitivity as a cutoff, 27.57% of unnecessary biopsies (which was significantly higher than 13.24% when using PSA) were avoided using UE-A, with only four (1.47%) csPCa patients being missed. AMACR combined with PCA3 has also been evaluated as a diagnostic marker of csPCa based on transcript detection in total urine RNA. A study by Kotova et al. [158] found a significant difference in the AMACR score between csPCa and nsPCa, in both the prebiopsy and pre-RP cohorts. Theoretically, UE-A may enhance decision-making regarding prostate biopsy alongside PSA, serving as another easy-to-obtain and non-invasive marker.

#### 3.2.6. Prostate-Specific Membrane Antigen (PSMA)

Prostate-specific membrane antigen (PSMA), which belongs to a class of integral membrane proteins first described in 1987, is a protein expressed in the cytoplasm of normal and malignant prostate tissue, including metastatic specimens [159]. In addition to PCa, the role of PSMA as a biomarker has been studied in other malignant tumors, such as renal cell carcinoma [160] and glioblastoma [161]. PSMA expression is significantly higher in primary PCa than in benign tissue and in distant metastases and metastatic lymph nodes than in primary tumors [162]. PSMA has a well-established, significant application in the detection, management, and follow-up of PCa patients. For example, in the case of biochemical recurrence after radical treatment, 68Ga-PSMA PET/CT can be performed to detect potential local recurrence or distant metastases [163]. It is important to note that the accuracy of this test depends on the level of PSA (detection rates around 50% with PSA levels ≤ 0.5 ng/mL and around 90% with PSA levels > 3 ng/mL) [164]. Additionally, during radio-guided salvage surgery, the intraoperative use of 99mTc-PSMA via a gamma probe may result in increased detection of metastatic sites [165]. PSMA expression is heterogeneous and has been found, through immunohistochemistry, to be negative in approximately 10% of primary PCa cases [166]. A recent study found that negative PSMA expression in the primary lesion was associated with the absence of PSMA expression in metastasis sites in a patient with a castration-resistant disease [167]. Another recent study found that PSMA-negative PCa was associated with negative PSMA-PET scans –even in patients with very high PSA levels. Importantly, this finding may influence the post-treatment surveillance of PSMA-negative patients and potentially help avoid negative PET-PSMA scans [168]. The association of PSMA overexpression in PCa tissues after radical prostatectomy with a poor biochemical recurrence-free survival rate and higher Gleason score has been established [169]. A study by Hupe et al. assessed PSMA expression in prostate biopsy samples. It was found that high PSMA expression was an independent prognostic marker in biopsy at the time of initial diagnosis. The five-year recurrence-free survival rates were 88.2% and 26.8% for patients with no and high PSMA expression on biopsy, respectively [170]. In another study, PSMA was used as one of five markers in a panel used to assess PCa and normal prostate tissue. The results showed that using a panel of several biomarkers (including PSMA) improved the detection of PCa compared with using each marker alone—especially in the case of the low expression of p504s, where the assessment was facilitated by positive PSMA [171]. Similar studies using immunohistochemical staining of PSMA in PCa tissue have found improved detection of aggressive PCa and prediction of therapy failure [172,173]. Further, more sensitive detection of PCa bone metastases was achieved with the use of PSMA (with NKX3.1) than of PSA [174]. Another very promising field is the use of PSMA as a target for targeted radionuclide therapy, especially alpha-particle radiation therapy. In targeted alpha therapy, radionuclides are used that release extremely energetic alpha particles chelated to monoclonal antibodies or small molecules designed to bind to PSMA. Although preclinical studies have demonstrated the potential effectiveness of this therapy, especially in metastatic PCa, further research is still needed to develop better methods for the synthesis of alpha-emitters and to then test these in clinical trials [175,176,177]. In a novel study by Allelein et al., a potential process for detecting PSMA in the urine of PCa patients was described. The researchers isolated prostate-cancer-derived extracellular vesicles from urine and assessed the value of PSMA as a biomarker for detecting PCa in a urine sample. To automate the whole process, they designed a device that detects PSMA-positive extracellular vesicles in urine. The authors concluded that the automated isolation of extracellular vesicles was feasible, but further research was needed before the described method could be used to detect PCa [178].

#### 3.2.7. Appl1

The adaptor protein containing a pleckstrin homology domain, phosphotyrosine binding domain, and leucine zipper motif 1 (Appl1) is one of the proteins localized in early endosomes [179,180]. Appl1 is considered a potentially important factor in the pathogenesis of PCa due to its multifunctional nature. To date, many of Appl1′s mechanisms of action have been described. Some of them, such as in controlling the speed of intracellular transport [181] or regulating transcription factors in the Wnt signaling pathway [182,183], are involved in the development and progression of PCa. Further, Appl1 acts through cytokine transforming growth factor-β type I receptor (TβRI), and the Appl1–TβRI complex was found to be associated with more aggressive PCa [184]. In a study by Martini et al., Appl1 was used as one of three immunohistochemical biomarkers on PCa tissues. Intense Appl1 labeling was observed in basal epithelial cells and in poorly formed malignant glands. The biomarker panel containing Appl1, Sortilin, and Syndacan-1 shows great value in improving the pathology assessment of prostate tissue (especially in distinguishing between benign and malignant tissue and in precise Gleason score classification) [185]. Appl1 alone may be used as a marker of aggressive PCa, potentially affecting the prognosis and follow-up patterns of patients treated for PCa. Additionally, it may be used in combination with other markers to support pathological examination and, in the future, improve the process of diagnosing PCa.

#### 3.2.8. Sortilin

Sortilin is a vacuolar protein sorting 10 family members that controls the transport of specific cargo within cells between structures, such as the Golgi apparatus, lysosomes, endosomes, and the plasma membrane [186]. It is particularly important in sugar metabolism and is highly expressed in tissues and cells with high energy demands. When stimulated by insulin, Sortilin helps facilitate transport to the plasma membrane by inducing the formation of vesicles and binding to glucose transporter 4 (GLUT4) and GLUT1 [187,188]. In PCa tissue, Sortilin is found in a granular pattern around the nuclei of cells and is more highly expressed in PCa cells with well-formed glands. Its ability to interact with both GLUT1 and GLUT4 suggests that Sortilin plays an important role in the metabolism of sugars, which, combined with Sortilin’s significant expression in PCa cells, suggests the dependence of these cells on sugar metabolism. In the previously mentioned study, Sortilin was one member of the biomarker panel that showed effectiveness in improving the pathological diagnosis of PCa—especially in the differentiation of benign and malignant lesions—by labeling only well-formed malignant glands with specific supranuclear polar patterns [185]. A study by Tanimoto et al. found that Sortilin, by promoting progranulin degeneration, might contribute to delaying the progression of castration-resistant prostate cancer [189]. More research is needed to unambiguously determine the role of Sortilin in PCa pathogenesis as well as its clinical implications.

#### 3.2.9. Syndecan-1

Syndecan-1, also known as CD138, is a transmembrane proteoglycan involved in various processes, such as cell proliferation, migration, and interactions with the extracellular matrix [190,191]. In PCa, Syndecan-1 expression has been observed in tissue samples with advanced cancer morphologies, including poorly formed glands, nests and cords of cells, and cribriform and intraductal carcinoma patterns. Its expression has also been detected in the tissues of patients with higher Gleason grades. Syndecan-1 has a direct role in binding growth factors and promoting cell migration, and its expression has been linked to the biochemical recurrence of PCa after radical prostatectomy [192]. In a previously cited study by Martini et al., Syndecan-1 was used as a marker and, along with two other previously described proteins (Appl1 and Sortilin), showed its usefulness in assessing histopathological prostate samples. Specifically, the use of Syndecan-1 epitope in the study provided reliable labeling of PCa with poorly formed gland morphologies, suggesting that it might have functional significance and potential utility for assessing advanced PCa using immunohistochemistry techniques [185]. A different study showed that cytoplasmic Syndecan-1 immunostaining was a predictor of poor prognosis in PCa and was associated with a higher Gleason grade and higher tumor stage as well as the occurrence of nodal metastases [193]. In [194], Santos et al. studied the expression of the Syndecan family in PCa tissues, finding that the overexpression of Syndecan-1 (as well as Syndecan-3) was associated with more aggressive tumors and a worse prognosis, and more precisely with decreased recurrence-free survival. In contrast, the overexpression of Syndecan-4 was correlated with a better prognosis [194]. Further, another study investigating the clinical implications of circulating Syndecan-1 in PCa patients found high levels (>123 ng/mL) of soluble Syndecan-1 in the serum of more advanced PCa patients, which was correlated with worse overall and cancer-specific survival. The authors suggested that the evaluation of soluble Syndecan-1 levels may be a promising tool for risk-stratification and therapy monitoring [195]. Another study [196] found that high levels of Syndecan-1 in serum were correlated with worse prostate-cancer-specific survival; additionally, they were found to be correlation with shorter survival in CRPC patients treated with docetaxel [196]. Syndecan-1 is a promising marker with many potential applications in diagnosing, treating, and monitoring PCa patients. However, to date, too few studies have been conducted to clearly indicate its clinical use, although the currently available research results are encouraging.

#### 3.2.10. p63

The last biomarker to be described is p63, also known as tumor protein. p63 belongs to the p53 protein family and is known for its role in tumor suppression [197]. It is located in the basal cells of many epithelial tissues and appears to be essential for proper gland development [198]. Upregulation of p63 has been linked with poor prognosis and cancer aggressiveness in ovarian carcinoma [199] and oral squamous cell carcinoma [200]. In contrast to the previously described markers, p63 is mainly used as an additional immunomarker alongside HMW-CK for distinguishing benign prostate hyperplasia from PCa [201]. The differentiation of prostate lesions on the sole basis of morphologic findings is sometimes challenging. Therefore, basal cells, which express HMW-CK and p63, are widely used in differentiating benign glands (in which basal cells are present) from PCa glands (in which basal cells are absent) [202,203]. While HMW-CK staining seems to be more sensitive than p63 in basal cell identification (90.70% vs. 88.37%, *p* = 0.7) [201], p63 still serves as a complementary marker in difficult cases [204]. A study by Kalantari MR et al. [205] stated that although both HMW-CK and p63 presented high specificity and sensitivity in distinguishing true adenocarcinoma from benign lesions (BPH), p63 seemed to be more specific than HMW-CK in distinguishing PCa mimickers (such as adenosis, atrophy, and partial atrophy [206]) from adenocarcinoma. Although nuclear p63 staining, which is a negative marker for neoplasm, is usually observed in normal basal cells, there is a subset of PCa cancer cells for which cytoplasmic p63 staining is found [207,208]. Interestingly, a study found that elevated expression of cytoplasmic p63 is associated with prostate-cancer-specific mortality, reduced levels of apoptosis, and increased cellular proliferation (for which there were high levels of ki67 protein, a proliferation marker [209]). This association persisted after multivariable adjustment for age, year of diagnosis, Gleason score, and stage [210]. Interestingly, a similar correlation has been observed for lung carcinoma [211]. This nucleus to the cytoplasm mislocalization of the p63 protein, which regulates proliferation and apoptosis, is suggestive of its potential oncogenic function.

**Table 1 biomedicines-11-01552-t001:** Summary of the characteristics of novel histopathological biomarkers described in this review.

Marker	Localization	Function	Clinical Implications
CD169	Extracellular; surface antigen on macrophages	Tumor immunity [3].	Improved survival [3,4].
Neuropilin-1 (NRP1)	Extracellular; transmembrane co-receptor	Vascularization and progression of cancers [6].	Higher Gleason and T scores. Positive nodal status. Progression to mCRPC [11].
CD15	Extracellular	Adhesion of cancer cells to the blood vessel endothelium [29,30,31,32]. Changes the structure of prostate cells mucins—NK cells cannot detect cancer cells [35].	Disease aggressiveness. Hormone-refractory type of PCa [24,28].
Cofilin-1	Intracellular	Monomer binding; cytoskeleton reorganization [67].	Higher Gleason score [73]. Positive nodal status [74]. Chemoresistance [67,68].
Signal transducer and activator of transcription protein 3 (STAT3)	Intracellular	Transcription activator; promotes tumor cell proliferation [75,83].	Worse overall and disease-free survival. Progression to hormone-refractory type of PCa [94].
LIMK1	Intracellular	Reorganization of actin cytoskeleton; intracellular androgen receptor signaling [120,121,122,123].	Independent risk factor for lymph node metastasis and biochemical recurrence after prostatectomy [128].
IL-17 family	Intracellular	Induces and mediates proinflammatory responses [42,43].	Unclear, elevated levels of IL-17RC in CRPC [60,61]; higher expression of IL-17F in cancers with higher histological grades [66].
AMACR	Intracellular; localized in peroxisomes and mitochondria	Degradation of branched fatty acids [133].	Negative marker for benignity of prostate glands [139]. Possible role as a novel non-invasive diagnostic marker [153].
Prostate-specific membrane antigen (PSMA)	Intracellular	Integral membrane protein [155].	Detection of recurrence or metastases in PET/CT [159]. Radio-guided salvage surgery [161]. High expression linked with poor prognosis [166].
Appl1	Intracellular; in early endosomes	Controls intracellular transport speed [174].	Associated with more aggressive PCa [177]. Improves the pathology diagnosis and grade of PCa [178].
Sortilin	Intracellular	Intracellular transport; involved in sugar metabolism [179,180].	Improves the pathology diagnosis and grade of PCa [178]. Delays progression of CRPC [182].
Syndecan-1	Intracellular/transmembrane	Cell proliferation, migration [183,184].	Improves the pathology diagnosis and grade of PCa [178]. Predictor of poor prognosis [186].
p63	Intracellular; usually nuclei of basal cells	Epithelium development, regulation of proliferation and apoptosis [191].	Positive marker for benignity of prostate glands [194].

PCa: prostate cancer; mCRPC: metastatic castration-resistant prostate cancer; CRPC: castration-resistant prostate cancer.

## 4. New Perspectives for Biomarkers in PCa

Since the population of patients with PCa is increasing worldwide, it is becoming increasingly important to detect all patients with clinically significant PCa, especially those at risk of metastasis, and to either apply adequate primary treatment or offer the most beneficial adjuvant therapy. Some new biomarkers may be very helpful during the primary diagnosis of PCa, with some studies already having indicated their usefulness in the assessment of biopsy material by a histopathologist (Appl1, Sortilin, Syndecan-1, p63), while others have indicated that they may be used in the non-invasive diagnosis of PCa (AMACR, PSMA) in the future. Not only have the roles of the presented markers been demonstrated in laboratory conditions, but they also have real clinical implications. For instance, they have been found to be correlated with increased resistance to chemotherapy and higher Gleason scores in tissue samples (cofilin-1, IL-17), worse oncological outcomes (STAT3, LIMK1, PSMA), improved survival (CD169), and progression to mCRPC (CD15, STAT3, neuropilin-1).

The different mechanisms of action of extra- and intracellular markers are important in the process of forming the pre-metastatic niche, the pro-cancerous environment of future metastasis sites. These particular biomarkers may help with the early detection of PCa in both the subclinical and more advanced metastatic stages and can serve as possible targets for new anticancer drugs.

A promising branch of new anticancer agents includes poly(ADP-ribose) polymerase (PARP) inhibitors, which seem to be effective in PCa patients with germline BRCA2 mutation [212]. Thanks to extensive sequencing efforts, the genomic environment of PCa is now better understood. DNA damage response (DDR) pathways preserve genomic stability by monitoring DNA integrity, activating the DNA repair process, or, if required, causing cell death. Out of all germline and somatic DDR mutations, the most commonly altered gene is BRCA2 [213]. Germline BRCA2 mutation was found to be a negative prognostic factor on CSS in mCRPC patients (17.4 months vs. 33.2 months in non-carriers) [214]. The inactivation of DDR genes affects 19% of patients with localized prostate tumors, and defects in the BRCA2 gene have been identified in 3–5.3% of patients [215,216]. The PARP family of enzymes plays an important role in DNA repair in a process called PARylation, which marks DNA lesions for repair. Targeting the PARP enzyme using specific inhibitors results in the accumulation of chromosomal instability, cell cycle arrest, and subsequent apoptosis [217]. Currently, ongoing trials evaluate the efficacy of olaparib, rucaparib, niraparib, and talazoparib, either as a monotherapy or part of combination therapy for prostate cancer in different disease stages. The first results suggest a particular response in patients with BRCA1/2 mutations compared to patients with non-BRCA mutations [212].

Despite the lack of high-quality studies on the expression of CD169 macrophages in prostate tissue, we have decided to propose its use as a biomarker due to its potential impact on the pre-metastatic niche in lymph nodes. To date, research focused on the detection of these macrophages in lymph nodes has found that low concentrations of CD169 macrophages are correlated with shorter PCa survival time. Based on the results of studies on CD169 macrophage expression conducted directly on tissues of other urological [5] and non-urological cancers [3], we believe that future studies on the expression of this marker in PCa tissues will also demonstrate its usefulness—especially in predicting the risk of lymph node involvement. In our opinion, any factor that helps to assess the risk of lymph node metastases is valuable, given that the currently available modalities for PCa patients are still imperfect.

Today, the diagnosis of PCa dissemination is based mostly on lymph node examination in post-RP material, yet lymphadenectomy is burdened with complications and does not bring survival benefits. The side effects of extended lymph node dissection include a longer operating time, increased blood loss, longer length of stay, and postoperative complications such as lymphocele [218]. However, to date, there is no better tool to determine the staging and prognosis of patients, and pelvic node dissection should be performed in intermediate- and high-risk PCa patients. The other traditional risk factors of biochemical recurrence after RP include PSA, PSAD, BMI, pathological Gleason score, seminal vesicle invasion, extraprostatic extension, and intraductal carcinoma [219]. The sentinel lymph node technique (SLN) may improve the detection of metastatic disease while limiting the possible complications resulting from extended lymphadenectomy. Despite promising results, its use remains controversial, and it is not recommended as a standard procedure [220,221]. To improve the intraoperative detection of sentinel nodes or other potentially metastatic lymph nodes, new techniques using gamma probes or cameras with fluorescence detection have been used. The use of radiotracers such as 99mTc-PSMA [165], indocyanine green (ICG)-99mTc-nanocolloid, or ICG alone followed by intraoperative fluorescence detection increases the detection of sentinel lymph nodes, and with the development of probe miniaturization technology, it is possible to use them more extensively during minimally invasive operations such as robotic or laparoscopic prostatectomy [222,223]. It is noteworthy that the results of a study by Claps et al. showed that the use of ICG alone allowed for the correct assessment of lymph node status in 98% of patients undergoing surgery. In addition, the authors emphasize the high negative predictive value of this method and suggest that if fluorescence is not detected during the intraoperative assessment, lymphadenectomy can be safely omitted, which significantly reduces the possibility of previously described complications [224].

Another still intensively developed and promising technique aimed at improving the diagnosis and prognosis of prostate cancer (especially in terms of the risk of metastases) is the so-called liquid biopsy (LB). It is a minimally invasive method that detects, for example, circulating cells or the genetic material of prostate cancer or other metabolites associated with active cancer disease in the patient’s blood (or urine). The advantage of LB is the ease of collecting material for testing, which can be repeated many times if necessary without significant harm to the patient [225]. A detailed description of this method and the currently studied markers detected during LB goes well beyond the scope of this article. However, considering the promising results of research on some of the detected markers, we would like to point out a few of them. The first is circulating tumor DNA (ctDNA) with a very wide potential application, from supplementing the currently used diagnostic methods by facilitating risk group stratification before treatment to monitoring the effectiveness of treatment and prognosis of the course of prostate cancer (e.g., castration-resistance development) [226]. The next biomarker is a group of microRNAs (miRNA) consisting of many different subtypes (such as miRNA-21 and miRNA-141) that show significant potential, especially in terms of predicting the occurrence of metastases or castration resistance. Other biomarkers include bone metabolites, such as bone sialoprotein (high levels are associated with a shorter time to bone metastasis formation) and osteopontin (can be used to assess treatment response in patients with crPCa) [227]. In addition to the promising results, the authors emphasize that the LB technique still requires improvement and the development of standards regarding, among others, the collection of samples for testing and their appropriate storage or the detection methods of the proposed biomarkers. An important aspect is the high cost of using this method [226].

A promising idea for improving patient prognosis is the evaluation of additional specific PCa markers in pre-RP biopsy specimens. More effective assessment of the biopsy material, correct differentiation between benign and malignant lesions, and the accurate assessment of the Gleason score by a histopathologist are key initiating steps in the diagnostic and therapeutic path of a patient with PCa. Biomarkers such as Appl1, Sortilin, Syndecan-1, and p63 have already proven their usefulness in this process.

In the future, these biomarkers could be used not only as helpful prognostic factors for patients after radical prostatectomy but also serve as tools for selecting the most beneficial adjuvant therapy; for instance, patients with a high expression of neuropilin-1 are better candidates for adjuvant chemotherapy than for hormonotherapy.

New biomarkers may become targets of the rapidly developing targeted radionuclide therapy, as shown in the example of PSMA. This potentially represents a big step in the further development of new adjuvant therapies for the most advanced PCa patients, giving them a chance at longer survival or greater control of their symptoms, but more clinical studies are required.

Probably, the most promising aspect of the abovementioned markers is their potential use as targets for new immunotherapeutic drugs. Snuderl et al. [17] found that NRP1 plays a major role in the spread of medulloblastoma tumor cells. Targeting the neuropilin-1 receptor resulted in tumor regression, decreased metastasis, and improved survival of mice. Although this study was performed on a murine model and not on PCa, which is our focus, its results are promising. In that study, LIMK1 inhibition also resulted in reduced PCa cell motility and increased cell apoptosis. The inhibition of LIMK1 also affected AR-dependent cell processes and, in the future, could be used in pharmaceuticals, possibly alongside docetaxel or abiraterone in hormone-resistant PCa patients.

Other markers have also been investigated for their use as potential targets in the systemic treatment of prostate cancer, but such research is scarce and in its early stages. For example, the effectiveness of docetaxel in prostate cancer cells was found to be greater when there is reduced expression of CFL1 [71,72]. These results suggest a benefit may be derived from the combined use of immunotherapy and chemotherapy in the treatment of prostate cancer. The results of numerous studies [93,94,95,96,97] on the therapeutic potential of STAT3 blockers are very promising, but there are currently no such studies focusing on prostate cancer. For example, the results of the first in-human trial [94] examining the effects of STAT3 blockade in head and neck cancers are optimistic, suggesting the possible development of new therapies for cancers insensitive to currently used systemic treatments. A study [42] evaluating the effect of thieno [2,3-b] pyridine anticancer compound on breast and prostate cancer stem cells found a significant reduction in the number of CD15s-positive prostate cancer cells after therapy, but these are preliminary—the first results of such studies. Further research is required to evaluate the potential use of this therapy in the treatment of prostate cancer. Currently, there are no studies that have evaluated the use of IL-17 and AMACR as targets of new anticancer drugs, but researchers that have described their mechanisms of action have emphasized their potential use in the development of new therapies (for example, blocking the IL-17–MMP7–EMT axis [69] or inhibiting AMACR’s conversion of CRPC to hormone-dependent PCa [155,156]).

Nevertheless, there is a need to conduct further research in order to find new drugs that can target specific tissue markers in PCa patients.

It is also worth noting that new technologies such as artificial intelligence (AI) and machine learning (ML) are emerging as powerful tools in the field of prostate cancer diagnosis and management as well as in the analysis of histopathological samples [228,229]. ML algorithms, a subset of AI, can be trained on large datasets of medical images and clinical data to help clinicians make more accurate and informed decisions regarding treatment options for prostate cancer patients. In recent years, AI-based algorithms have shown promising results in detecting prostate cancer from medical images, including multiparametric magnetic resonance imaging (mpMRI) (e.g., distinguishing between benign and cancer prostate tissue [230] or predicting Gleason score [231,232]) and ultrasound (e.g., detecting high-grade prostate cancer [233]). Additionally, AI-based systems are being developed to analyze histopathological samples of prostate cancer, which may help pathologists make more accurate and reliable diagnoses. Issues such as the classification of biopsy samples using the Gleason scoring system [234,235,236] and the transformation of examined images into three dimensions in order to increase detection sensitivity and improve tissue material evaluation [237] are just some of the areas where the use of artificial intelligence has provided promising, but still preliminary, results. Currently, the use of ML cannot adequately replace a specialized histopathologist. However, in the future, it is expected to significantly improve their work and thus shorten the time needed to evaluate tissue material and increase the accuracy of their expert conclusions. Introducing new immunohistochemical markers to protocols for the histopathological evaluation of prostate tissues may provide new, valuable data for AI analysis, which may further increase the effectiveness of evaluation. In addition, ML can be used as an additional tool to predict patient mortality and the risk of prostate cancer recurrence based on various clinical, imaging, and histopathological data [238,239,240]. By combining machine learning with traditional diagnostic methods, AI has the potential to improve the accuracy and efficiency of prostate cancer diagnosis, management, and treatment, ultimately leading to improved patient outcomes. However, more research is needed to fully evaluate the potential benefits and limitations of AI in the context of prostate cancer.

## 5. Conclusions

To summarize, novel histopathological biomarkers may facilitate the decision-making process with regard to patients with PCa. To date, there have been too few studies on these biomarkers to conclusively assess their clinical application; however, the early results are very promising, especially regarding their use in the assessment of the risk of metastasis or recurrence after radical treatment. More quality research is required to incorporate these markers into clinical practice.

## Figures and Tables

**Figure 1 biomedicines-11-01552-f001:**
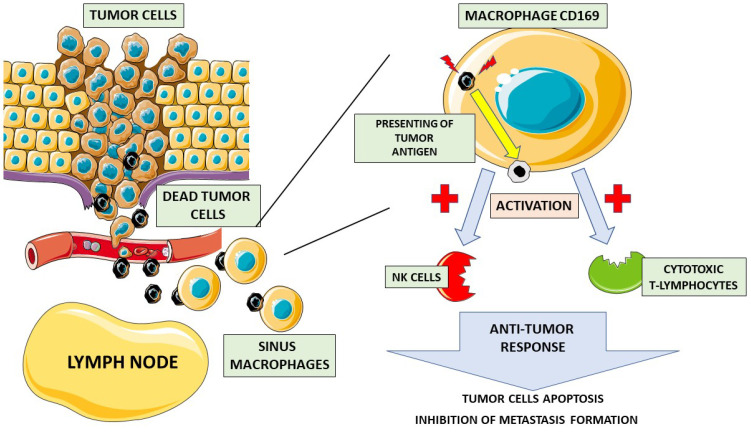
Role of CD169 macrophages in anticancer response. Dead cancer cells travel through the bloodstream to the lymph nodes, where they are phagocytized by CD169 sinus macrophages. Then, CD169 macrophages present cancer cell antigens on their surface, activating natural killer cells (NK cells) and cytotoxic T-lymphocytes. Upon activation, an anti-tumor response takes place, leading to cancer cell apoptosis and inhibiting the metastatic process.

**Figure 2 biomedicines-11-01552-f002:**
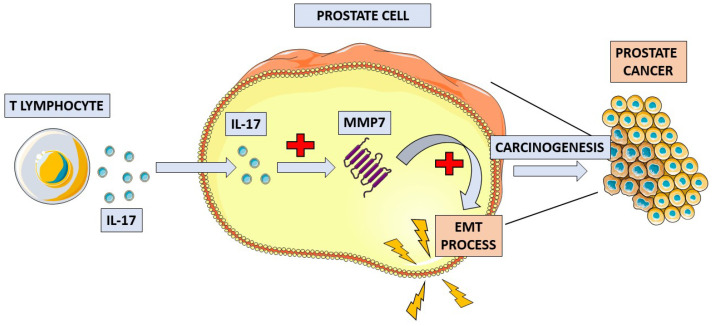
Schematic overview of the proposed mechanism of IL-17 in prostate cancer pathogenesis. IL-17 increases the concentration of matrix metalloproteinase 7 (MMP7) in prostate tissue. MMP7 induces the process of epithelial-to-mesenchymal transition (EMT), which leads to the development of prostate cancer.

## Data Availability

Not applicable.

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
