# Peer review of "Novel Histopathological Biomarkers in Prostate Cancer: Implications and Perspectives"

_biomedicines, 2023, doi:10.3390/biomedicines11061552_

Round 1
Reviewer 1 Report
This review paper "Novel histopathological biomarkers in prostate cancer: implications and perspectives" addresses clinically relevant biomarkers in the context of disease progression as well as overall survival of patients with prostate cancer.
One of the strengths of the paper is that the topic and every biomarker is presented in a detail manner, the paper is appropriately structured and sufficiently refers to recent studies.
The entire manuscript is written in easy-to-read scientific language. The role of further studies to increase the impact of these biomarkers on the clinical management of prostate cancer patients was also discussed.
Reviewer 2 Report
The authors made a thorough compilation of the studies performed to advance the field of histological prostate cancer biomarkers. They provided a clear picture on how the field has moved forward and the potential steps for the future application of these biomarkers. The work is well-written and easy to follow. Just a few comments:
Line 258: The authors could use "breast tumors" instead of "breast cancer tissue".
Line 348: Use "cancer types" instead of "cancer diseases".
Line 401: Remove "in". It should read: In addition to PCa...
Reviewer 3 Report
In this MS, authors reviewed histopathological biomarkers and their potential application in decision making regarding the prognosis and treatment of prostate cancers. However, it has some concerns as follows:
1. Due to novelty issue, show rationale of this review compared to previous similar reviews.
1) Prostate Cancer Biomarkers: From diagnosis to prognosis and precision-guided therapeutics 2) An overview of biomarkers in the diagnosis and management of prostate cancer 3) Biomarkers in prostate cancer - Current clinical utility and future perspectives 4) Urinary biomarkers of prostate cancer
2) Most biomarkers were reported as PCa diagnostic biomarkers. That’s why I am wondering whether “novel” can be used in title.
Reviewer 4 Report
In this manuscript the authors provided a comprehensive review of the literature about the current role and future perspectives of histopathological biomarkers in prostate cancer. The paper provides some important insights on the topic, the effort is to be commended.
For a more clinical perspective the authors should add an additive paragraph about the clinical implications of novel histopath. biomarkers in the context of intraoperative analysis of sentinel lymph nodes retrieved by PSMA-, Tc-, ICG-fluorescence guided pelvic lymph node dissection as an hot and emerging topic of the current urological debate. There is the lack of reliable technique for intraoperative lymph node analysis and the authors should consider to further discuss these aspects. The authors should consider to refer to (doi: 10.1007/s11701-022-01382-0; doi: 10.1016/j.urolonc.2022.08.005; doi: 10.2967/jnumed.120.259788; doi: 10.1111/iju.14513; doi: 10.1016/j.eururo.2020.10.031). This paragraph would be very interesting for the readers. The decision to abort or proceed to PLND during RALP is a crucial step of the radical treatment.
The authors should further add a "Methods" section in which briefly describe the timeframe and the settings of the search strategy.
Reviewer 5 Report
In their study, the authors summarize histo-pathological markers, which in the future may help to establish the prognosis of patients with PCa as well as assist in choosing the optimal adjuvant therapy. The manuscript is straightforward, well written, and concise and has clear results within the scope of a review article. Definitely deserves to be published and is a valuable contribution to the “Biomedicines” journal. The following comments should be addressed before publication, based on my recommendations.
[1] “1. Introduction”, Page 2 of 26, Lines 46-48:
“Tissue markers represent tumor heterogeneity, which in clinical practice denotes different responses to certain types of adjuvant therapy, which we will further describe in this review.”.
At that point, the authors should report that urinary liquid biopsy is attractive and promising for PCa detection. Apart from the specific biomarkers from urine, potential serum biomarkers that may allow the precision medicine revolution to take place include androgen receptor variants, bone metabolism, neuroendocrine and metabolite biomarkers. Furthermore, in the subset of patients with bone metastases, bone sialoprotein (BSP) and osteopontin (OPN) have demonstrated prognostic value. Higher BSP levels are related to a shorter time to develop bone metastases in patients with PCa. OPN may be of use in assessing treatment response post chemotherapy in patients with castration-resistant PCa.
Recommended reference: Saxby H, et al. An Update on the Prognostic and Predictive Serum Biomarkers in Metastatic Prostate Cancer. Diagnostics (Basel). 2020;10(8):549.
[2] “2.1.2. Neuropilin-1 (NRP1)”, Page 2 of 26, Lines 108-110:
“Creating a specific environment for tumor progression requires vascularization, which in turn relies on VEGF, for which NRP1 is one of the receptors crucial for vessel development.”.
The authors are strongly recommended to discuss the correlation between HIF-1 and VEGF. HIF-1 is composed of an alpha and a beta subunit (HIF-1a and HIF-b). HIF-1a is hydroxylated by HIF prolyl-hydroxylase, which then targets HIF-1a for degradation under normoxic conditions. Hydroxylated HIF-1a is specifically ubiquitinated by the von Hippel-Lindau E3 ubiquitin ligase, marking HIF-1a for proteasomal degeneration. Under hypoxic circumstances, the hydroxylation of HIF-1a is restricted by the disposal of oxygen molecules and HIF-1a is secured and assembles. HIF-1a can then dichotomize with HIF-b and prompt the transcription of hypoxia-survival genes. Among the transcripts managed by HIF-1 is VEGF.
Recommended reference: Ioannidou E, et al. Angiogenesis and Anti-Angiogenic Treatment in Prostate Cancer: Mechanisms of Action and Molecular Targets. Int J Mol Sci. 2021;22(18):9926.
[3] “3. New perspectives for biomarkers in PCa”, Page 11 of 26, Lines 563-565:
“These particular biomarkers may help with the early detection of PCa in both the subclinical and more advanced metastatic stages and can serve as possible targets for new anticancer drugs.”.
The authors should mention that large-scale sequencing efforts have allowed a better understanding of the genomic landscape of PC. Beyond maintenance of androgen receptor signaling in a castration setting, alternative mechanisms have been proposed for disease progression. In particular, germline or somatic aberrations in the DNA damage repair genes are found in 19% of primary PC and almost 23% of metastatic castration-resistant PC and compromise genomic integrity. As such, several PARP inhibitors have been investigated in metastatic castration-resistant PC patients and are therapeutically effective in germline BRCA2 mutants.
Recommended reference: Shah S, et al. BRCA Mutations in Prostate Cancer: Assessment, Implications and Treatment Considerations. Int J Mol Sci. 2021;22(23):12628.
